# A high-resolution perspective of extreme rainfall and river flow under extreme climate change in Southeast Asia

Mugni Hadi Hariadi[1,2,3], Gerard van der Schrier[1], Gert-Jan Steeneveld[2], Samuel J. Sutanto[4],
Edwin Sutanudjaja[5], Dian Nur Ratri[3], Ardhasena Sopaheluwakan[3], and Albert Klein Tank[6]

[1]Royal Netherlands Meteorological Institute (KNMI), De Bilt, Netherlands
[2]Meteorology and Air Quality, Wageningen University and Research (WUR), Wageningen, Netherlands
[3]Indonesian Agency for Meteorology, Climatology and Geophysics (BMKG), Jakarta, Indonesia
[4]Earth Systems and Global Change, Wageningen University and Research (WUR), Wageningen, Netherlands
[5]Utrecht University, Utrecht, Netherlands
[6]Met Office Hadley Centre for Climate Science and Services, Exeter, United Kingdom

**Correspondence:** Mugni Hadi Hariadi. Royal Netherlands Meteorological Institute (KNMI) Utrechtseweg 297, 3731 GA De
Bilt (Email: mugni.hariadi@knmi.nl; mugni.hariadi@bmkg.go.id; mugnihadi@gmail.com)

**Abstract.** This article provides high-resolution information on the projected changes in annual extreme rainfall and high and
low streamflow events over Southeast Asia under extreme climate change. The analysis was performed using the bias-corrected
result of the High-Resolution Model Intercomparison Project (HighResMIP) multi-model experiment for the period 1971-2050.
Eleven rainfall indices were calculated along with streamflow simulation using the PCR-GLOBWB hydrological model. The

historical period 1981-2010 and the near-future period 2021-2050 were considered for this analysis. Results indicate that over
Indochina, Myanmar will face more challenges in the near future. The east coast of Myanmar will experience more extreme
high rainfall conditions, while northern Myanmar will have longer dry spells. Over the Indonesian maritime continent, Sumatra
and Java will suffer from an increase in dry spell length of up to 40%, while the increase of extreme high rainfall will occur over
Borneo and mountainous areas in Papua. Based on the streamflow analysis, the impact of climate change is more prominent

in a low flow event than in a high flow event. The majority of rivers in the central Mekong catchment, Sumatra, Peninsular
Malaysia, Borneo, and Java will experience more extreme low flow events. More extreme dry conditions in the near future
are also seen from the increasing probability of future low flow occurrences, which reaches 101% and 90% on average over
Sumatra and Java, respectively. In addition, based on our results over Java and Sumatra, we found that the changes in extreme
high and low streamflow events are more pronounced in rivers with steep hydrographs (rivers where flash floods are easily

triggered), while rivers with flat hydrographs have a higher risk in the probability of low flow change.

## 1 Introduction

The IPCC's sixth assessment report (IPCC AR6) indicates that Southeast Asia (SEA) is one of the most vulnerable regions to
climatic changes and thus is highly exposed to the impacts of climate change (IPCC, 2021). Smit and Wandel (2006) explained
that vulnerability is usually considered to be the product of three elements: exposure, sensitivity, and adaptive capacity. One of

the challenges SEA faces is that climate change leads to increasing extreme rainfall, and this condition is aggravated by the low

resilience and adaptive capacity of most developing countries in SEA (Hijioka et al., 2014). Beyond extreme rainfall events, climate change in SEA manifests in various detrimental forms, including sea level rise, drought, high temperatures, and the resulting increasing scarcity of freshwater, biodiversity loss, and other facets of environmental degradation, all of which exert negative impacts on Southeast Asian economies and societies Jasparro and Taylor (2008). The climate change vulnerability mapping for SEA has been documented by Yusuf and Francisco (2009). Those extreme events, especially extreme rainfall, are projected to intensify in the future under climate warming (Ali and Mishra, 2017).

Several previous studies documented the observed changes in climate in the SEA region (Supari et al., 2017; Siswanto et al., 2016; Suhaila et al., 2010; Cinco et al., 2014). Supari et al. (2017) found a tendency towards wetter conditions by looking at the simple daily precipitation intensity (SDII), a significantly increasing trend in the annual highest daily rainfall amount (RX1day), and the rainfall amount contributed by the extremely very wet days (R99p). In addition, Siswanto et al. (2016) observed trends in extreme rainfall over Jakarta - Indonesia and found that the number of days with rainfall exceeding 50 and 100 mm per day shows a statistically significant increase from 1961 to 2010. They added that the trends in extremes are strongest during the wet season compared to the dry season. For Peninsular Malaysia, Suhaila et al. (2010) studied trend patterns during the wet and dry seasons and they found a decrease (increase) in total rainfall and a significant decrease (increase) in the frequency of wet days leading to a significant increase in rainfall intensity during the southwest (northeast) monsoon over most of the region. Moreover, In Thailand, Limsakul et al. (2010) revealed changes in rainfall extreme events along Thailand's coastal zones in recent decades observed from three different stations; Andaman Sea Coast, the Gulf of Thailand's western coast, and the Gulf of Thailand's eastern coast. These authors found an overall decrease in total rainfall amounts accompanied by a coherent reduction of heavy and intense rainfall events in the Andaman Sea Coast, and more intense daily rainfall associated with a significant decrease in the number of rainy days in the Gulf of Thailand's western coast. However, in the Gulf of Thailand's eastern coast, the changes in extreme precipitation were relatively mixed between significant positive and negative trends. In addition, Cinco et al. (2014) observed an increase in extreme events over 34 synoptic weather stations in the Philippines for the period 1951-2010 compared to the normal mean values for the period 1961-1990.

The variations and changes in precipitation extremes in SEA become a crucial factor for several sectors of life, e.g., the agricultural sector when addressing food security issues (Knox et al., 2012; Lin et al., 2022; Redfern et al., 2012), economics (Weiss et al., 2009), and the hydrological sector (Hoang et al., 2016). Hydrology plays an important role in meeting the grand challenges of many sectors, such as availability of fresh water, food security, supply of water to hydropower facilities, the vitality of ecosystems, sanitation and sustainable development (Singh and Xiaosheng, 2019). A state-of-the-art hydrological model becomes an essential tool for the effective planning and management of water resources and recently, research has broadened to include sustainable water in relation to climate change-induced changing patterns of streamflow. A decrease in streamflow for such a long period may cause drought that can trigger impacts on the water supply, energy, water-borne transportation, and ecosystems (Stahl et al., 2016). Studying this can help decision-makers in managing the river basin as a key factor affecting the volume of water required to cope with the increasing demands of the population and several specific activities such as agriculture, energy, and tourism sectors, which directly depend on water resources (Mair and Fares, 2010).

Given that precipitation is the key factor influencing streamflow and hydrological response in catchments (Lobligeois et al., 2014), an approach centred on assessing changes in both precipitation and hydrology becomes imperative. Climate indices are simple diagnostic quantities that can be used to describe the state changes in the climate system. The examples are the many impact-relevant indices to measure precipitation changes and variation from the Expert Team on Climate Change Detection Indices (ETCCDI) (Klein Tank et al., 2009). For scientific and operational purposes, exploring the space-time variability of

hydrologic extremes in relation to climate is important (Renard and Thyer, 2019). Renard and Thyer (2019) study shows that climate indices have frequently been used as predictors to describe hydrologic extremes.

    Although the impact of climate change on rainfall and hydroclimatic extremes in SEA has been extensively studied, most of these studies have relied on data from the previous version of the climate model, in this case, the Coupled Model Inter-comparison Project Phase 5 (CMIP5). The regional climate model (RCM) output based on CMIP5 (CORDEX-SEA) has been

used in prior research, revealing an increasing risk of droughts and extreme rainfall events across SEA (Ngai et al., 2020a, b; Nguyen-Ngoc-Bich et al., 2021; Supari et al., 2020; Tangang et al., 2018, 2019; Trinh-Tuan et al., 2019). Some studies within this dataset have focused on specific rivers in Malaysia to show the impact of climate change on hydro-meteorological droughts (Tan et al., 2019, 2020). The limited use of the latest version of high-resolution CMIP (CMIP6) on this subject is attributed to the unavailability of high-resolution data from CMIP6 RCM for SEA at the current time. This presents a challenge, particu-

larly considering that the CMIP6 model has been demonstrated to offer a superior representation of many processes (physical, chemical, and biological) compared to the CMIP5 model (Li et al., 2021; Zhu and Yang, 2020). Previous studies have indicated that over SEA, the CMIP6 high-resolution Modeling Intercomparison Project (HighResMIP) (Haarsma et al., 2016) closely simulated monsoon characteristics (Hariadi et al., 2021) and rainfall indices (Hariadi et al., 2022) to observation, outperforming CORDEX-SEA. Furthermore, Hariadi et al. (2021) observed that the atmospheric-only experiment of HighResMIP success-

fully replicated deviations in monsoon characteristics observed during El Niño years. Thus we have relatively high confidence in the HighResMIP model results.

    Therefore, this study aims to quantify the change of river flow under changing extreme rainfall due to the near future climate change over SEA, using HighResMIP model results (Haarsma et al., 2016) as the meteorological driver. A bias-corrected version of the HighResMIP is constructed to document the changes in the rainfall-related climate indices, in this case; rainfall

indices and investigate the change and the trend of the indices. This dataset is then used to simulate the river streamflow over four domains in the SEA region; the Mekong basin (MEB), the Sumatra-Peninsular Malaysia domain (SMB), the island of Java, and Borneo. Furthermore, the changes in streamflow values during low flow and high flow events and their probability in the near future are investigated.

    Section 2 (Materials and Methods) of this study describes the study area, the climate and streamflow data used, and methods

that include climate indices, the bias correction method and the statistical methods for the high and low flow indices. Furthermore, in section 3 (Results), we present the change in climate indices and streamflow over SEA in the near future period (2021-2050) compared to the historical period (1981-2010). Section 4 (Discussion) compares our findings to previous studies on the change of climate indices. In addition, we also discuss the impact of the change in climate indices, the impact of catch-

ment properties on the hydrological extremes, and the source of uncertainty related to the result. We conclude our findings in section 5.

## 2 Materials and Methods

### 2.1 Description of the study area

The Southeast Asia domain (SEA) used in this study is located between 14.8°S - 34°N and 89.5°E - 146.5°E. This includes the Indonesia maritime region in the South, the Philippines maritime region in the East, and Indochina (South of China, Myanmar, Thailand, Laos, Cambodia, Vietnam, and peninsular Malaysia) in the Northwest. This domain covers the Mekong basin in the North SEA (Fig. 1). The mountain region over Southern China to northern Laos is part of the upstream part of the Mekong rivers. On the maritime continents, mountain ranges are spread from the Philippines to islands in Indonesia, such as Sumatra, Java, Sulawesi, and Papua. These mountains specify the streamflow characteristics in the regions.

The rainfall in these regions is dominated by the monsoon season which starts from the North SEA in May and moves to the South SEA in November (Aldrian and Susanto, 2003; Hamada et al., 2002; Hariadi et al., 2021; Moron et al., 2009). Some areas in the SEA, such as Myanmar (Li et al., 2013), Vietnam (Nguyen-Thi et al., 2012; Luu et al., 2021) and Philippine (Corporal-Lodangco and Leslie, 2017) are affected by tropical cyclones, which lead to high extreme precipitation.

### 2.2 Data

#### 2.2.1 Climate data

Our study uses the climate model output from the high-resolution model intercomparison project (HighResMIP) (Haarsma et al., 2016) that is available from the H2020-funded Primavera project (Roberts et al., 2020). This model has a spatial resolution (25-50km spatial resolution) comparable to the downscaled output of the regional climate model (RCM) (25km spatial resolution) which is based on CMIP5 over SEA (CORDEX-SEA). Previous studies show that the HighResMIP has a better simulation of the monsoon characteristics (Hariadi et al., 2021) and extreme precipitation (Hariadi et al., 2022) than the CORDEX simulations over SEA when compared against the observational SA-OBS (Van den Besselaar et al., 2017), APHRODITE (Yatagai et al., 2012), and CHIRPS (Funk et al., 2015) observational datasets. We use the coupled atmospheric-ocean model historic runs of the HighResMIP for the historical period 1950-2014 (Hist-1950) and the future period 2014-2050 (highres-future). Compared to the historically forced atmosphere run with the prescribed sea surface temperature (SST) version of HighResMIP (HighResSST), the Hist-1950 version shows similar performance on simulating the monsoon characteristic (Hariadi et al., 2021) and the extreme precipitation (Hariadi et al., 2022) over SEA, which shows the skill of the ocean model in the coupled models. We used five models that are available from the Hist-1950, which are the CMCC (Cherchi et al., 2019), CNRM (Voldoire et al., 2019), EC-Earth (Haarsma et al., 2020), HadGEM (Roberts et al., 2019) and MPI (Müller et al., 2018). Only one member is available for the CMCC, CNRM and MPI model simulations, while EC-Earth and HadGEM have four and three members available.

## 2.2.2 Streamflow data

Global hydrological models (GHMs) have been developed over the last decade and become essential tools to quantify the global water cycle. The GHMs simulate distributed hydrological responses to climate and weather variations at a higher resolution than in general circulation models (GCMs). One of the recently developed GHMs is a grid-based global hydrological model called PCR-GLOBWB (PCRaster Global Water Balance) (Van Beek et al., 2011; van Beek et al., 2012). PCR-GLOBWB describes the terrestrial part of the hydrological cycle that focuses on global water availability issues (Van Beek et al., 2011; van Beek et al., 2012). Sutanudjaja et al. (2018) added more advanced run-off processes, river routing, and groundwater components to this model i.e., PCR-GLOBWB 2.0 and extensively evaluated its performance.

We used the PCR-GLOBWB 2.0 (Sutanudjaja et al., 2018) in this study to simulate historical and future streamflows. PCR-GLOBWB is essentially a leaky bucket type of model (Bergstrom, 1995) applied on a cell-by-cell basis. For each grid cell, PCR-GLOBWB calculates the daily water storage in two vertically stacked soil layers (max. depth 0.3 and 1.2 m) and an underlying groundwater layer, as well as the water exchange between the layers and between the top layer and the atmosphere (rainfall, evaporation and snow melt) (Van Beek and Bierkens, 2009). The modelled terrestrial water balance then provides the runoff which is used for the streamflow modelling. Sutanudjaja et al. (2018) described the parameters, the standard input data, and the parameterisation of PCR-GLOBWB 2.0 which is mostly the same as the one for the preceding version PCR-GLOBWB 1.0 (Bierkens and Van Beek, 2009). There are 5 modules in the PCR-GLOBWB 2.0: the meteorological forcing, the land surface, the groundwater, the surface water routing, and the irrigation and water use that are calculated in a daily time step (Ruijsch et al., 2021). In this study, we use the configuration as in Sutanudjaja et al. (2018). The difference is that in our study we exclude the 'water used' factor and focus more on meteorological exposure. The motivation is that it is beyond the scope of this paper to assess and include future changes in water use. The model runs in 5 arcmins spatial resolution, which is about 10 km by 10 km at the equator. We simulated daily water discharge for the period of 1971-2050, based on bias-corrected model data using the PCR-GLOBWB hydrological model. The First-order conservative remapping method was used to interpolate the bias-corrected climate data models into 5 arcmin's spatial resolution. Rainfall and the air temperature were used as input, while the potential evapotranspiration was estimated using the Hamon method (Hamon, 1961) which is available in the PCR-GLOBWB. To simulate a better fluctuation of daily streamflow, we selected the kinematic wave for the routing method, which allows flow and area to vary both spatially and temporally within a conduit. Thus PCR-GLOBWB simulates the river discharge in $^3$/s for all river networks.

The simulation was run for four domains, they are the Mekong Basin (MEB), the Sumatra-Malaysia Basin (SMB), and the islands of Java and Borneo. Sutanudjaja et al. (2018) validated the PCR-GLOBWB 2.0 simulation with streamflow data from the Global Runoff Data Centre (GRDC). The forcing data set for the simulation is based on time series of monthly precipitation, temperature, and reference evaporation from the CRU TS 3.2 (Harris et al., 2014) downscaled to daily values with ERA40 (1958–1978) (Uppala et al., 2005) and ERA-Interim (1979–2015) (Dee et al., 2011). Their result shows the model correlation and Kling-Gupta Efficiency coefficient or KGE (Gupta et al., 2009) values range from 0.21 to 0.98 and from -6.49 to 0.87 for MEB (137 observation sites), 0.29 to 0.70 and -2.51 to 0.41 for SMB (14 observation sites), 0.16 to 0.80 and

-1,98 to 0.34 for Java island (10 observation sites), and 0.36 to 0.70 and 0.07 to 0.34 for Borneo island (5 observation sites),
respectively. This indicates that the model performs well enough to simulate the observed streamflow in many river basins
across SEA. In addition, the PCR-GLOBWB model is proven to be reliable for studies on extreme streamflow (Van der Wiel
et al., 2019; Yossef et al., 2012). In their study, Van der Wiel et al. (2019) utilize the PCR-GLOBWB hydrological model and
the EC-Earth global climate model as input. They assess the return period of an extreme hydrological event by conducting
a 2000-year simulation of global hydrology under both present-day and 2°C warmer climate conditions. Furthermore, Meng
et al. (2020) utilize PCR-GLOBWB to analyze the future hydropower production under 1.5°C than 2°C climate scenario over
Sumatra.

### 2.3 Methods

#### 2.3.1 Climate Indices

In this study, we used 11 rainfall-related climate indices, which were earlier used to assess the realism of model simulations
(Hariadi et al., 2022). The indices were calculated using the package developed by Schulzweida and Quast (2015), which is
part of the climate data operator (CDO) suite of routines (Schulzweida et al., 2006). Nine indices were adopted from the Expert
Team on Climate Change Detection Indices (ETCCDI) team (Klein Tank et al., 2009). In addition, we also calculated the
number of consecutive dry days periods (CDD) and consecutive wet days periods (CWD) exceeding five days (CDD5D and
CWD5D) that are available in the package (Schulzweida and Quast, 2015). The indices are aggregated to the annual level. The
percentile value for the rainfall fraction due to very wet days (exceeding the 95[th] percentile) (R95pTOT) is calculated based on
the 1971-2050 period. Table 1 lists names and definitions of the rainfall-related climate indices computed in this study.

#### 2.3.2 Bias Correction

The global circulation model is handicapped by biases to the degree that prevents their direct use for hydrological purposes
(Ehret et al., 2012). Earlier studies discussed the biases in the CMIP5 model simulation (e.g., Taylor et al., 2012), where the
bias is worse for precipitation simulation over regions with complex topography (Mehran et al., 2014). Ngai et al. (2017)
discussed the need for bias correction on precipitation and temperature simulation of regional climate models over SEA.

Empirical quantile mapping (EQM) is one of the bias correction methods that is widely used. The number of quantiles in this
method is a free parameter Piani et al. (2010). Previous studies used the EQM as a quantile-quantile calibration method based on
a nonparametric function that corrects biases in the cumulative distribution functions (CDFs) of climatic variables (Boé et al.,
2007; Amengual et al., 2012). Recently, Fang et al. (2015) compared bias correction methods in downscaling meteorological
variables for a hydrologic impact study in China, Ratri et al. (2019) used EQM to bias-corrected ECMWF SEAS5 over Java
island, Indonesia, and Hariadi (2017); Ngai et al. (2017); Amsal et al. (2019) used quantile mapping to bias-corrected RCM
simulation over SEA. A recent study by Ngai et al. (2022) also used quantile mapping (QM) to explore the possible ranges
of future rainfall and extreme index changes over SEA. They noted that the QM method modifies the climate change signal
by expanding the range of change when correcting for biases in future projections even though the impact of the QM bias

correction is different for RCMs with different indices. Overall, the QM bias correction slightly increases the magnitude of projection change, and the strongest effects (either magnification or reduction) mostly happened in Indochina (Ngai et al., 2022). For example, during December-February (DJF), the effects on the magnitude of change correspond well to the mean rainfall distribution in SEA. During June-August (JJA), the effects can be found in both Indochina and the Maritime Continent in ensemble-mean and in some models, especially for the changes of seasonal rainfall over Java - Indonesia and the southern region (Ngai et al., 2022; Tangang et al., 2020). They also showed that the values of high rainfall change slightly increased mostly in the northern part and in the Indochina mainland. Moreover, it is possible that the direction of rainfall changes after QM bias correction.

EQM works with empirical probability density functions (PDFs) or CDFs for forecasts and observations. This method attempts to correct the distribution of the GCM and RCM simulated data so that it matches the distribution of the observational dataset (Déqué et al., 2007; Block et al., 2009). EQM estimates quantiles for the forecast and the observation dataset and forms a transfer function by using corresponding quantile values. Then, each predicted quantile is substituted by the corresponding observed quantile using their Empirical CDFs (ECDFs). The transfer function is then applied to the forecast data as follows.

$$Y_{f(bc)} = ECDF_o^{-1}[ECDF_f(Y_f)]$$

where $Y_f$ is the raw precipitation forecast, and $Y_{f(bc)}$ is the bias-corrected precipitation re-forecast. $ECDF_o$ is the inverse ECDF of the observations, and $ECDF_f$ is the ECDF of the forecast values.

We used the rainfall and temperature gridded dataset obtained from the APHRODITE (Yatagai et al., 2012) as the observation (reference data) in the bias correction. The reference period for the bias correction is 1971-2010. For rainfall, the APHRODITE V1101 for the period of 1971-2006 was combined with the APHRODITE V1101 EXR1 for the period of 2007-2010. Whereas for the temperature, we use APHRODITE V1808. Limited gauge density and availability of long-term climatological data make the development of a dataset for daily precipitation amounts based on in-situ measurements challenging (Van den Besselaar et al., 2017; Singh and Xiaosheng, 2019). The Southeast Asia Observation dataset (SA-OBS) (Van den Besselaar et al., 2017) developed especially for SEA has the highest density of gauges than other datasets available. However, limited coverage of SA-OBS does not cover the entire Mekong basin. This is the reason we used APHRODITE for this study. Hariadi et al. (2022) found that both APHRODITE and SA-OBS that developed from gauge data have more similarities than between both datasets with the Climate Hazards Group Infrared Precipitation with Stations v2.0 (CHIRPS; Funk et al., 2015) which is based on satellite data.

Figure 2 and S1 show the results from the two-sample Kolmogorov–Smirnov test (K-S) of the original and bias-corrected model. It shows that the K-S statistic is lower for the bias-corrected simulation compared to the original (uncorrected) model. This indicates that the probability distribution of the bias-corrected model is closer to the observations compared to the uncorrected model for simulating these climate indices. More improvements are shown in the model simulation of climate indices that are directly related to rainfall intensity (R10mm, R20mm, Rx1day, Rx5day, R5day50mm, and SDII) than other indices that are more climatological (CDD, CDD5D, CWD, CWD5D, and R95pTOT). Hariadi et al. (2022) also found that the model

poorly simulates climate indices that are directly related to rainfall intensity. Based on the K-S value, we find a significant improvement in the model simulation on these climate indices after the bias correction was performed. Here, we show the importance of the bias-correction process for model simulation on climate indices. This study, therefore, uses the bias-corrected dataset for further analysis.

### 2.3.3 High and low flow indices

The high and low flow indices were identified using the threshold-based approach. This approach applies the theory of runs and is developed based on a pre-defined threshold level for each index (Yevjevich, 1967; Hisdal et al., 2004; Sutanto and Van Lanen, 2021). Thresholds in this study were derived from the $10^{th}$ and the $95^{th}$ percentiles of the daily streamflow (Q10 and Q95 of flow duration curve), which are the flows that are either equal or lower than 10 percent of the time or exceeded 95 percent of the time. We calculated both the $10^{th}$ and the $95^{th}$ percentiles of the daily discharge for the combined historical and the near future periods. The $10^{th}$ percentile represents low flow discharge (LFD) (Tallaksen et al., 1997; Wong et al., 2011), while the $95^{th}$ percentile identifies high flow discharge (HFD) (van Vliet et al., 2013; Asadieh and Krakauer, 2017). We investigate the change of LFD and HFD in the near future period (2021-2050) compared to the historical period (1981-2010). A decrease in LFD indicates that the driest 10% of daily discharges are drier than those for the historical period, whereas an increase in HFD indicates more severity of the 5% most extreme high discharge events. Using the $10^{th}$ and $95^{th}$ percentiles of the historical period as a reference of extreme events, we calculate the probability change of the extreme low and extreme high events in the near future. The probability change is calculated based on the change of the number of events that exceeded extreme low and extreme high reference values in the future compared to the historical. For example, if the occurrence of future extreme is doubled than the historical period, then the probability increase is 100%. The increase in the probability indicates that events considered extreme in the historical period will occur more frequently in the near future.

## 3 Results

### 3.1 Change in climate indices over Southeast Asia

Climate indices (Table 1) for the period 1971-2050, derived from the bias-corrected models, were calculated to show the change in extreme rainfall. We calculated the change of climate indices between the near-future period (NF, 2021-2050) and the historical period (Hist, 1981-2010). In addition, we also calculated the trend over the period 1971-2050 including the Sens slope significance test with a 95% confidence level (Sen, 1968). The final change in the climate index values is based on the model mean. Acknowledging uncertainties, where extreme values from certain models are included in the averaging process, a trend significance test was also performed. This test is based on the model agreement and remains unaffected by extreme values, as they are excluded from the trend analysis. A trend is considered significant when 3 or more out of 5 models (60%) show a significant result. For EC-Earth and HadGEM, which have 4 and 3 members respectively, the trend is considered significant when at least 3 and 2 members show a significant trend.

Figure 3a shows increasing CDD over some areas in the Philippines, the northern part of Myanmar, the southern part of Vietnam and Thailand, Cambodia, and the centre of Peninsular Malaysia. Over the Indonesian region, an increase between 20% to 40% is seen in the Southern part of Sumatra. The increase of CDD is also found in Java, Bali, Nusa Tenggara, the Southern part of Borneo, and the Northern part of Sulawesi. Similar to the change in Indochina and the Philippines, the increase is less than 20% in most of these areas. In addition, there is also a decrease of CDD in the mountains region of Papua up to 15%. We found a robust signal of increasing CDD over the northern part of Myanmar, the southern part of Sumatra, and some areas in Java which not only show the increasing change of CDD value but also show a significant trend up to 3 days/decade (Fig. 3a). The drier condition in the near future over the Southern part of Sumatera is also shown by the increasing CDD5D over the region, with a significant increasing trend of CDD5D (Supplementary Material Fig. S2). Most of the areas may experience higher CDD and CDD5D in SEA also show a decrease in CWD and CWD5D (Supplementary Material Fig S3 and S4).

The change in frequency of heavy rainfall events (precipitation events with daily amounts greater than or equal to 20 mm (R20mm)) in the near future compared to the historical period is also apparent in our result (Figure 3a). Northwestern Indochina and several islands in the maritime continents, such as Sumatra, Borneo, Sulawesi, and Papua are projected to have an increase of R20mm. A high increase (>40%) of R20mm is found over some areas in the Eastern part of Borneo and mountainous areas in the northern part of Papua. Based on a model agreement, a significant increasing trend for both R20mm and R10mm also appears over Borneo and the mountainous area in northern Papua (Fig. 3b and S5b). This clearly indicates a robust signal of increasing heavy and very heavy rainfall events over those regions.

The change in intensity of yearly maximum one-day precipitation (Rx1day) is depicted in Figure 3c. Rx1day shows an increasing intensity scattered over Indochina, especially in the west coast and the northern part of Myanmar, the west and east coast of Peninsular Malaysia, and some areas in Thailand, Cambodia, and the southern part of Vietnam. Over the Indonesian region, Rx1day increases over Borneo, Sumatra, Sulawesi, and mountainous areas over Papua. A similar pattern is also found for the maximum 5-day precipitation (Rx5day) as shown in Supplementary Figure S6a. Although Rx1day and Rx5day exhibit increasing trends spreading over SEA, the trends are stronger over the west coast of Myanmar and the mountainous area of Papua (Fig. 3c and S6b).

The change in the rainfall fraction due to extreme wet days (exceeding the 95th percentile) (R95pTOT) is shown in Figure 3d. The figures clearly indicate an increase of R95pTOT across SEA. In the near future, large percentages of areas with an increase in R95pTOT are found over west northern, and central Indochina, Malaysia peninsula, Sumatra, Borneo, Sulawesi, and Papua. Especially over the west coast of Myanmar, Borneo, and Papua, there is a high increase of R95pTOT (>20%). In terms of the model agreement on the trend significance, a significant increase in trend is found over west northern Indochina, Borneo, and the mountainous region in Papua.

Regarding the Simple Daily Intensity Index (SDII), Figure 3e shows an increase of SDII (>10%) over the western part of Myanmar, the east coast of Peninsular Malaysia, northern Philippines, Borneo, and Papua. A significant increasing trend of SDII is seen over the northern Philippines, Southern Sumatra, Sulawesi, Borneo, Papua, and some areas in Indochina. Over

Malaysia peninsula, a significant positive trend is found over the west coast of the Malaysia peninsula instead of the east coast of the region, which has a higher increasing value of SDII.

Mean and extreme rainfall exhibit seasonally dependent patterns. Supplementary material Figures 8-11 present the seasonal changes in rainfall indices (CDD, R20mm, Rx1day and R95pTOT). The analysis considers four seasonal periods: December to February (DJF), March to May (MAM), June to August (JJA), and September to November (SON). Results indicate that the projected increase in CDD in the near future is more pronounced during the periods of JJA and SON. This is particularly evident over the southern part of Sumatra, the southern part of Borneo, and the northern part of Myanmar (Fig. S8).

Figure 4 illustrates the seasonal changes in SDII in the near future. Results indicate an increase in SDII during DJF and MAM in the Indonesia region, suggesting an increase in rainfall intensity during the peak (DJF) and the end phase (MAM) of the monsoonal season. On the contrary, decreasing intensity (SDII) is observed over Indochina and the Philippines for these periods, especially for MAM. While, during JJA and SON, most of the SEA shows an increase in SDII, except for the southern part of Indonesia. Moreover, we also found a substantial increase in SDII over Vietnam during SON. This condition aligns with the patterns observed in R20mm, which shows a similar pattern (Fig. S9). Over the southern part of Indonesia (Java, Bali, and Nusa Tenggara), R20mm increases during DJF and MAM. In contrast, increasing SDII is noted over the Indochina region for SON and JJA. In addition, over the equatorial region (Sumatera, Borneo, and Papua), an increase in R20mm is observed for all periods. Similar conditions are also seen in Rx1day and R95pTOT (Fig. S10 and S11).

### 3.2 Change in streamflow over Southeast Asia

Streamflow is the volumetric flow rate of water per unit of time that is transported through a given river cross section. Figures 5a and 5b indicate the percentage of LFD change and the percentage of change in the probability of extreme low flow over MEB. Figures 5c and 5d show the percentage of HFD change and the percentage of change in the probability of extreme high flow over MEB. The decrease (increase) of LFD (HFD) indicates the more extreme condition of low (high) flow events in the near future. Figures 5 a and b show that future low flows will be more severe and the probability for low flows increases, these conditions are more general over the central and southern parts than over the northern part of MEB. Over 76% area of MEB will face a decrease in the magnitude of LFD (16% decrease on average), with 15% of this area showing a further reduction of more than 25% in discharge in the LFD events. In terms of low flow events, the largest part of the region (82%) shows an increasing probability for a Low Flow Discharge with a 66% increase (on average). A Significant decreasing LFD and increasing probability of low flow events appear in the centre part of MEB (25°N - 17°N), with 23% and 104% on average.

In terms of high discharge, half of the region shows an increase in the magnitude of HFD events (50% area) (Fig. 5c) and in the probability of high flow events (49% area) (Fig. 5d) in the near future. However, the magnitude for both indices is relatively low with only a 5% and 12% increase on average, respectively for HFD and probability of high flow events. Slightly in contrast to the low flows, more extreme conditions of high flow are found over the northern part of the region.

The percentage of LFD and HFD change and the probability change in extreme low and high flows in SMB are shown in Figure 6. Figures a and b show that future low flows will be more severe and the probability of low flows increases. Over 91% of SMB will see further reductions of flow with (on average) 19% but peaking area to average decreases of 22-24% for

the central (2°S - 2°N) and southern (below 2°S) parts Sumatra island. These are higher than the northern part of Sumatra and Peninsular Malaysia with (on average) 17% and 15% decreases. Furthermore, the probability of extreme low flow events will increase over 94% of SMB (101% on average), with 95% of these areas experiencing an increase in the probability of more than 25% compared to the historical (Fig. 6b). Over Sumatra, all parts of the island show increasing probability with (on average) 146%, 127%, and 76% increases for the northern, central, and southern parts, respectively. While in Peninsular

Malaysia, the average increase in the probability of extreme low flow events reaches 73%.

    In contrast to the low flows, the high flows will get less extreme (Fig. 6c) and the probability of high flows is insignificantly increasing (Fig. 6d) over the SBM region. In terms of high flow, 49% of the SMB region shows increasing HFD in the near future. On average, the increase in HFD magnitude occurs only over 7% across SMBs. Compared to other areas in the region, the increasing high discharge is more prominent in Peninsular Malaysia with 70% of this region showing a higher HFD of 10%

on average than in the past. In terms of High flow events, 59% of the region shows an increased probability of the events with an average 19% increase.

    In Java Island (Fig. 7), the increasing severity during low flow events in the near future is higher than the increasing severity during high flow events. The decreasing LFD magnitude will occur over a large area of Java (85%) (Fig. 7a). The average value of the decrease in LFD magnitude is 13%. Although the percentage area that shows decreasing LFD over the western

part (79%) is less compared to the eastern part (96%) of Java, the magnitude of the decreasing LFD is higher for the western part. The average decreasing LFD for the western and eastern parts of Java are 19% and 8%, respectively. A slightly different condition is shown in the probability of extreme low flow event compared to the low streamflow magnitude change over Java. The increasing probability of extreme low flow events is more pronounced in the eastern region compared to the western region. In the eastern region, 98% of the areas show an increasing probability, with an average increase of 126%. Meanwhile,

in the western region, 90% of the areas show an increasing probability, with an average increase of 52%. Overall, 95% of Java shows an increased probability of low follow events with a 90% increase on average (Fig. 7b). The increase of high flow magnitude over Java is lower than the corresponding change in low flow magnitude. 58% of Java shows an increasing HFD magnitude in the near future (Fig. 7c). However, the maximum increase of HFD in the Java island is only 11%, with 3% as an average. In general, the increasing HFD magnitude is relatively low in Java, unlike other locations in SEA. The increase

in the probability of extreme high flow events is more significant compared to the increase of HFD (Fig. 7d). A large part of Java (75%) will experience an increase in probability, with 25% of these areas indicating a higher probability of >25% in the future. The average of increasing probability is 16% across Java.

    In Borneo, both high and low flow events are projected to increase in the near future, especially in terms of the probability of change in extreme events. Here, even though it is projected that 91% of the area will experience lower LFD in the near future,

the magnitude is relatively low, which is only a 10% decrease in LFD on average (Figure 8a). The decreasing LFD over the southwest part of Borneo is more than in the other parts of the island. On the contrary with the relatively little decrease in low streamflow, the increase in the probability of extreme low events over Borneo is fairly significant. Figure 8b shows that the increase in the probability of low flow events will occur in 94% of Borneo Island, with 74% of these areas will experience a higher probability of >25%. In addition, the entire southern part of Borneo will experience an increasing probability of low

flow events, which is divided into the southwestern part (67% increase on average) and the southeastern part (59% increase on average).

Figure 8c shows a percentage of HFD changes in Borneo. It shows that most of the region (89%) will experience increasing HFD, spreading more from the northwest to the southeast. However, the increasing HFD is relatively low, with an average of 8%. Figure 8d demonstrates that the increase in the probability of extreme high flow events occurs over 90% of Borneo,

spreading more from the northwest to the southeast. The average increase in probability is 28%, with almost half of the increased probability being >25% in the near future.

## 4   Discussion

### 4.1   Extreme climatic changes

To understand past, present, and future climate changes, the Southeast Asia Regional Climate Downscaling/Coordinated

Regional Climate Downscaling Experiment (SEACLID/CORDEX–SEA) (http://www.ukm.edu.my/seaclid-cordex) group has conducted dynamic downscaling of a multi-model from CMIP5 into a high-resolution dataset (Cruz et al., 2017; Cruz et al., 2016; Cruz et al., 2017). In this sub-section, we compare our findings with previous studies from CORDEX-SEA. This study utilizes the HighResMIP model, which shares a similar resolution with the CORDEX-SEA but is based on the latest version of GCM (CMIP6). Furthermore, HIghResMIP was globally run on high resolution, while CORDEX-SEA resulted from a re-

gional climate model. The use of HIghResMIP in the study is expected to yield less uncertainty in the results compared to CORDEX-SEA (Hariadi et al., 2021, 2022; Tian-Jun et al., 2019).

Compared with CORDEX-SEA, we found some similarities and some differences in our results. Tangang et al. (2018) found that Myanmar will have more consequences of extreme events due to global warming than other regions in Indochina. Northern Myanmar will experience more severe extreme dry and wet conditions in the future under the 2°C global warming scenario.

This is confirmed in our results where we found that the northern part of this region shows an increasing dry spell length (CDD) and the models show an agreement on the increasing trend of CDD in the near future, indicating that this increase is a robust signal. In addition, the western coast of this region will also experience higher precipitation extremes, as indicated by some indices, such as Rx1day, Rx5day, R95pTOT, and SDII. The contrasting behaviour between longer dry periods and more intense rainfall is a trend that is observed globally as well (Benestad et al., 2022). Tangang et al. (2019) found that Thailand

will experience wetter conditions over the northern-central-eastern parts and drier conditions over the southern part. A similar pattern of index changes was also found in this study. However, based on the model agreement, we found no significant trend in these wet and dry conditions. Nguyen-Ngoc-Bich et al. (2021) found that based on the Palmer Drought Severity Index, substantial increases in drought duration, severity, and intensity appear over northern parts of the North Central sub-region, parts of the Central Highlands and over southern Vietnam. In our study, we could only confirm that longer dry spells (CDD)

appear over the east coast of the Southern part of Vietnam, but there is no model agreement on the trend significance.

Ngai et al. (2020a) demonstrated that rainfall extremes are likely to decrease (increase) over Peninsular Malaysia (Malaysian Borneo) by the end of the century. However, our result is contradicting in the sense that an increase in extreme rainfall is seen

over Peninsular Malaysia although the increase is less than in Malaysian Borneo. Some decreases in the number of heavy precipitation days (R10mm) are projected over some areas in Peninsular Malaysia but this decrease is not clear for other extreme rainfall indices. In addition, Ngai et al. (2020b) found high increases of R20mm, RX1day, R95pTOT, and R99pTOT, and also small increases in SDII and CDD over the Malaysia region. Our study yields similar conditions for those parameters except for SDII which shows a high increase. The increase of SDII spreads over the region and shows a significant trend over the western part of Peninsular Malaysia and the majority of Malaysian Borneo.

A previous study by Supari et al. (2020) found that the increase of CDD is higher than the increase of Rx1day and frequency of extremely heavy rainfall (>50mm) (R50mm), especially over the Indonesian region. Accordingly, Tangang et al. (2018) also simulated a robust increase in CDD, which indicates drier conditions over Indonesia in the future. The almost similar conditions between CDD and Rx1day are found in our study. Although the increase of Rx1day is scattered over SEA, the increasing CDD is more prominent than Rx1day, especially in the Indonesian region. In our study, we also calculated R20mm instead of R50mm, which slightly shows a different result. The increase of R20mm is more prominent than CDD. Over the Indonesia region, a significant increasing trend of R20mm appears over Borneo and the mountainous region in Papua. The CDD, on the other hand, has a significant positive trend over southern Sumatra and in Java. Moreover, the increase of CDD in Indonesia is higher than in other regions in SEA. This indicates that a condition toward dryer conditions is higher in the Indonesia region than in other regions in the SEA.

## 4.2   The impact of changes in climate indices on the hydrological extremes

In their investigation of climate change impacts on global streamflow, Van Vliet et al. (2013) observed changes in seasonal flow amplitudes, magnitude, and timing of high and low flow events. They projected that the mean flow and low flow will decline over SEA in the future. This is supported by other previous studies showing the streamflow change over the Johor (Tan et al., 2019) and Kelantan (Tan et al., 2020) river basins in Malaysia. In addition, they projected that the meteorological drought is likely to become longer at the end of the 21st century. However, it is still not clear whether the hydrological drought will have a longer duration. Although results from a study on meteorological drought - focusing on short-term precipitation shortage - not necessarily coincides with hydrological drought, low flow most likely will occur during an extreme drought event (Sutanto and Van Lanen, 2021). Similar to the global study conducted by Van Vliet et al. (2013), our research is in agreement with their finding by concluding that over SEA, the impact of extreme climate change in the near future is more prominent for the low flow than high flow conditions, especially in Indonesia.

The impact of changes in climate indices in SEA also affects the changes in hydrological extremes. The increase of CDD over Northern MEB (northern 24°N) results in declining LFD over the central part of MEB, especially over northern Laos. Our results also show that the increase of Rx1day and R95pTOT over the northern MEB does not significantly affect the HFD over the northern and central parts of MEB. However, the increase of Rx1day, R95pTOT, and SDII over the southern MEB yields higher HFD and the probability of high flow events over Thailand and Cambodia.

In the southern part of Sumatra and in the centre of Peninsular Malaysia, the CDD will increase in the near future, reducing the LFD across the region. This might be caused by the increase of CDD5D that spreads more than CDD. The increase of SDII

over the Southern part of Sumatra and the eastern part of Peninsular Malaysia results in higher streamflow and the probability of HFD in some rivers in these regions.

The increasing CDD that spreads in the majority of areas of Java makes most rivers on the island show a decrease in LFD. Increasing Rx1day, R95pTOT, and SDII over the north coast of the western part of the island lead to higher HFD and an increase in the probability of high-flow events.

In Borneo, the increase of CDD in the Southern part and CDD5D in most areas of the island decrease the LFD and increase the probability of low flow events. Although the increasing Rx1day is visible only in some areas in Borneo, most areas show higher R95pTOT and SDII. As a result, most of the rivers in Borneo have higher HFD and probability of high flow events.

Multiple sectors can be affected by the change in magnitude and probability of LFD. The likely impact of decreasing LFD over SEA will pose a challenge to some sectors such as agriculture, hydropower, industry, and public water supply. As mentioned by Horton et al. (2022), the decreasing flow in the early wet season in the future over the Cambodian Mekong floodplain will affect rice production. In addition, in Laos, hydropower will face challenges due to a 22% decrease in LFD and an 85% increase of probability low flow events in the central part of MEB. Meanwhile, over Sumatra, two big hydro-powered rivers will also face issues with decreasing LDF. The two hydropower sectors are firstly the Sigura-Gura hydropower located in the north part of Sumatra which will face a 15% LFD decrease and an 88% increasing probability of low flow events of the Asahan River. Secondly, in the western part, the Musi Bengkulu hydropower will face a 23% LFD decrease and a 75% increasing probability of low flow events of the Musi River. On another Island of Indonesia, Java, the Mrica hydropower on the Serayu River in the central part of the island will face a 17% LFD decrease and a 68% increasing probability of a low flow event. Over Borneo, the Kayan River will face an 8% LFD decrease and an 80% increasing probability of low flow events, this condition needs to be considered for the planning of the Kayan hydropower project which will be one of the biggest hydropower plants in SEA and will provide electricity for the new capital of Indonesia (Ibu Kota Nusantara - IKN). The increase in extreme dry rainfall conditions will decrease the LFD resulting in low water table in the soil. In Borneo, these conditions will decrease the peatland's water table, which will increase the risk of forest fires in that area and, for similar reasons, over Sumatra island (Taufik et al., 2017).

## 4.3 Impact of catchment properties on hydrological extremes

The impact of climate change varies between rivers depending on the catchment characteristics. This also applies to extreme events, such as floods, droughts, high flow, and low flow as shown in our study. Previous studies on drought demonstrated that hydrological extremes are not only influenced by climate but also by the catchment properties (Van Lanen et al., 2013; Van Loon and Laaha, 2015; Sutanto and Van Lanen, 2022). In our study, the role of the physical characteristics of the river can be clearly seen in the low flow analysis. To investigate the impact of catchment characteristics on the low flow, we calculated the river recession constant ($C$) (Gustard and Demuth, 2009). The $C$ value shows the overall recession rate in days, in which the small (high) $C$ indicates a steep (flat) hydrograph of the river (Gustard and Demuth, 2009). One should note that the recession constant ($C$) is not the same as the recession coefficient ($RC$). A steep hydrograph indicates that the river has a high risk of a

flash flood, while a flat hydrographic has a low risk of a flash flood. A description of how to calculate $C$ and the plots over four selected domains are available in supplementary material 12.

In MEB, increasing CDD over the northern part of the region has more impact on LFD over the central part of the basin. A similar result was also found in Java, where increasing CDD across the island has more impact on LFD for rivers flowing in the western part of the island. The $C$ analysis shows that these rivers have relatively small $C$. We found that rivers with small $C$ (steep hydrograph) are more susceptible to LFD changes than rivers with high $C$ (flat hydrograph), associated with catchment properties. Some interesting results are also found over Sumatra and Java regarding increasing the probability of low flow events. Although it did not show a relatively high decreasing LFD than other rivers in each domain, rivers in the Northern part of Sumatra, the Eastern part of Java, the southern part of Borneo, and also the eastern part of Borneo show a significant increase in the probability of low flow events. Those rivers are found to have a relatively high $C$ value.

Next, we discuss the relation between the change in low flow and catchment properties denoted by $C$ by plotting the density of the streamflow over three rivers for historical and near future periods. The plot of the density of the discharge over three rivers is available in supplement material 13. The Bengawan Solo River located in the eastern part of Java and the Kampar River located in the northern part of Sumatra are rivers that have high $C$ values. We compared those rivers with the Batanghari River which is situated in the southern part of Sumatra that has a small $C$ value. Here we found that rivers with high $C$ values have a more narrow streamflow distribution than rivers with small $C$ values. This is due to the more steady and less flashy streamflow characteristic of high $C$ value rivers. As a result, the shift of streamflow distribution for high $C$ value rivers triggers a higher probability of low flow change than for small $C$ value rivers.

## 4.4 Sources of uncertainty

There are three sources of uncertainty related to the result which are climate-forcing inputs, hydrological model structure, and hydrological model parameters. In terms of climate forcing, this study used one climate scenario (RCP 8.5) which is the projection based on a high future emissions scenario for temperature and greenhouse gasses (global warming scenario). The use of this single scenario becomes one of the sources of uncertainty in the results as it makes the results describe only one climate change scenario. However, IPCC (2021) reports relatively small differences among RCP scenarios in the near future (until 2050) climate projection. In addition, Lehner et al. (2020) reported that major uncertainty for the near future rainfall projection comes from the model spread instead of the RCP scenario itself. This indicates that the uncertainty of using one RCP scenario is low. Moreover, the RCP8.5 scenario has covered the extreme spectrum of climate projection. In order to reduce the uncertainty coming from the model spread, we employed 5 high-resolution GCMs (10 members in total).

The second source of uncertainty comes from the hydrological model structure. In this study, we applied only one hydrological model (PCR-GLOBWB), which is a limitation of this study. This study, however, focuses on the impacts of extreme climates on future high and low flows. The PCR-GLOBWB model is one of the global hydrological models that is run in high resolution globally and it is proven reliable for extreme studies (Van der Wiel et al., 2019; Yossef et al., 2012). In addition, the hydrological model simulation does not consider changes in water demand by agriculture or human consumption. This is

also one of the limitations of the near future flow analysis. The purpose of this setting is that we want to focus only on climate exposure on extreme streamflows.

490     The third source of uncertainty comes from the PCR-GLOBWB parameterization. One aspect of uncertainty is that the PCR-GLOBWB uses the Hamon method (Hamon, 1961) for the potential evaporation estimation. Temperature is the only input for the method and this might amplify the effect of changing temperature on the result. However, the actual evapotranspiration is limited by the availability of water in water-stressed conditions, which will make this effect smaller. In addition, parameterization of deep groundwater, soil layer, and land cover also contributes to the uncertainty of the simulated flow. Nevertheless, 495 Sutanudjaja et al. (2018) shows that the streamflow simulation is reliable for our study regions.

## 5   Conclusions

The SEA region will experience an increase in both dry and wet extremes in the near future. Myanmar and Peninsular Malaysia will face more challenges due to climate change compared to other areas in the Indochina region as the changes are the strongest in these regions. The northern part of Myanmar will experience an increase in dry spell length (CDD), while the west coast 500 of Myanmar will experience more extreme rainfall (in terms of the wettest day of the year, Rx1day, the wettest 5-day spell of the year, Rx5day, and the number of very wet days, R95pTOT). Peninsular Malaysia will also experience increasingly long dry spells (CDD, CDD5D), wet spells with excessive amounts of rain (Rx5day50mm) and an increase in the intensity of rainfall (SDII). The amount of rainfall due to very wet days in Malaysia (R95pTOT) will increase in the near future. Furthermore, increasing dry spell length (CDD) is also found in the Central part of the Philippines, along with an increase in 505 rainfall intensity in the northern Philippines. Over the Indonesian Maritime continents, Sumatra and Borneo will experience more extreme conditions of dry and wet events in the near future. The southern part of Sumatra and Java will be affected by the highest increase of dry spell length (CDD) up to 40% in the SEA. In addition, Sumatra, Borneo, and Papua will experience increasingly intense rainfall (SDII) and stronger rainfall extremes (R10mm, R20mm, Rx1day, R95pTOT).

    As a result of changing rainfall, we found decreasing and increasing low flow (LFD) and high flow (HFD), respectively along 510 with the increasing probability of low flow and high flow events in the near future. A drier condition during the low flow event is more prominent compared to a wetter condition during the high flow event. In the Mekong basin, the decrease in streamflow and increase in the probability of low flow events are found in the Central and Southern parts of the Basin, with higher change in the central part. In contrast, increasing discharge and probability of high flow events are projected in the southern part of the basin, especially in Thailand and Cambodia. Drier conditions during the low flow event and the increase in the probability of 515 low flow events were simulated in most of the rivers located in Peninsular Malaysia, Sumatra, Borneo, and Java. The increasing probability of future low flow events reaches 101% and 90% on average over Sumatra and Java, respectively. The increase of high flow and its probability is found in some rivers situated in Peninsular Malaysia, western Java, and the majority of rivers in Borneo. In general, rivers in Borneo will experience more severe conditions during both low flow and high flow events.

    Our study also concluded that rivers in Sumatra and Java that have a less steady and more variable or 'flashy' streamflow 520 (quantified as steep hydrographs) are likely to experience more decreasing low flow in the future than rivers with a flat hy-

drograph. However, these latter rivers will have a greater risk of increasing the probability of low flow events than the rivers with steep hydrographs. This is associated with the characteristic of flat hydrograph rivers that generate a narrow discharge distribution. Our study reveals that the changes in low flow events and their probabilities are not only influenced by extreme dry climates but also by the catchment characteristics.

*Code availability.* PCR-GLOBWB model is available in `https://github.com/UU-Hydro/PCR-GLOBWB_model`

*Data availability.* The climate model output from the high-resolution model intercomparison project (HighResMIP) is available from the H2020-funded Primavera project ($https : //www.primavera - h2020.eu/$) and the APHRODITE gridded precipitation and temperature datasets are available in $http : //aphrodite.st.hirosaki - u.ac.jp/products.html$

*Author contributions.* Gerard van der Schrier: Conceptualization; resources; supervision; review. Gert-Jan Steeneveld: Supervision; review. 530 Samuel J. Sutanto: Conceptualization; supervision; review. Edwin Sutanudjaja: Software. Dian Nur Ratri: Writing and editing. Ardhasena Sopaheluwakan: Supervision. Albert Klein Tank: Supervision; project administration.

*Competing interests.* The authors declare that they have no known competing interests or personal relationships that could have appeared to influence the work reported in this paper.

*Acknowledgements.* The First author received funding from the Indonesia Endowment Fund for Education (LPDP)(S-353/LPDP.3/2019) for 535 his PhD program. The second author acknowledges the support of the Royal Netherlands Embassy in Jakarta, Indonesia, through a Joint Cooperation Programme between Dutch and Indonesian research institutes. The HighResMIP simulations have been made available through the PRIMAVERA project which received funding from the European Union's Horizon 2020 Research and Innovation Programme under grant agreement no. 641727. This PRIMAVERA data is part of the IS-ENES3 project that has received funding from the European Union's Horizon 2020 research and innovation programme under grant agreement No. 824084.

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

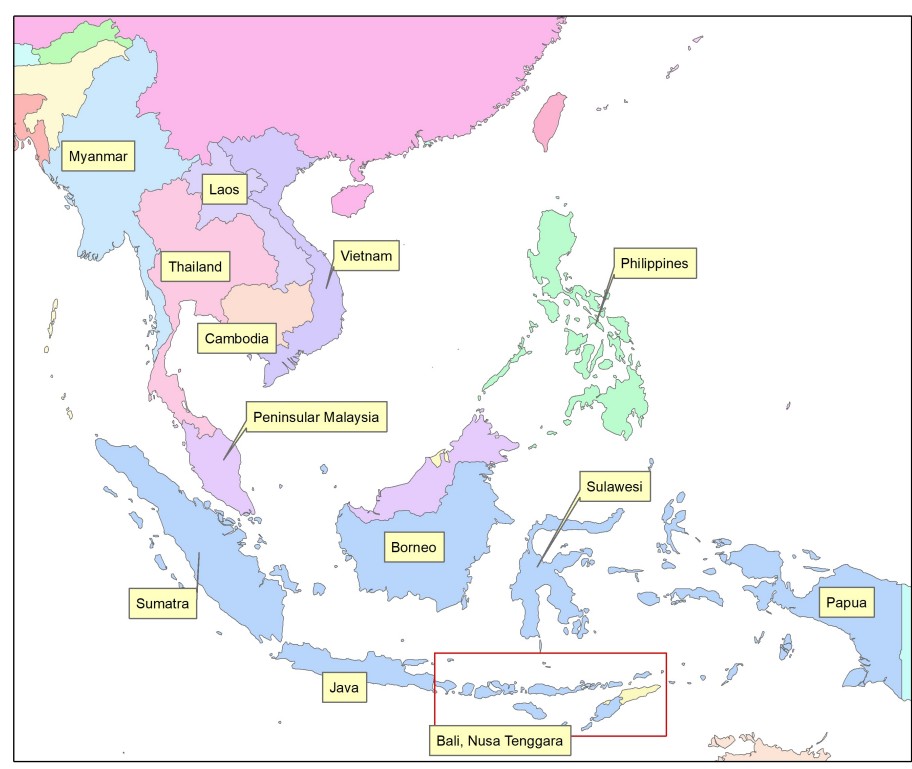

**Figure 1.** The Southeast Asia domain in this study

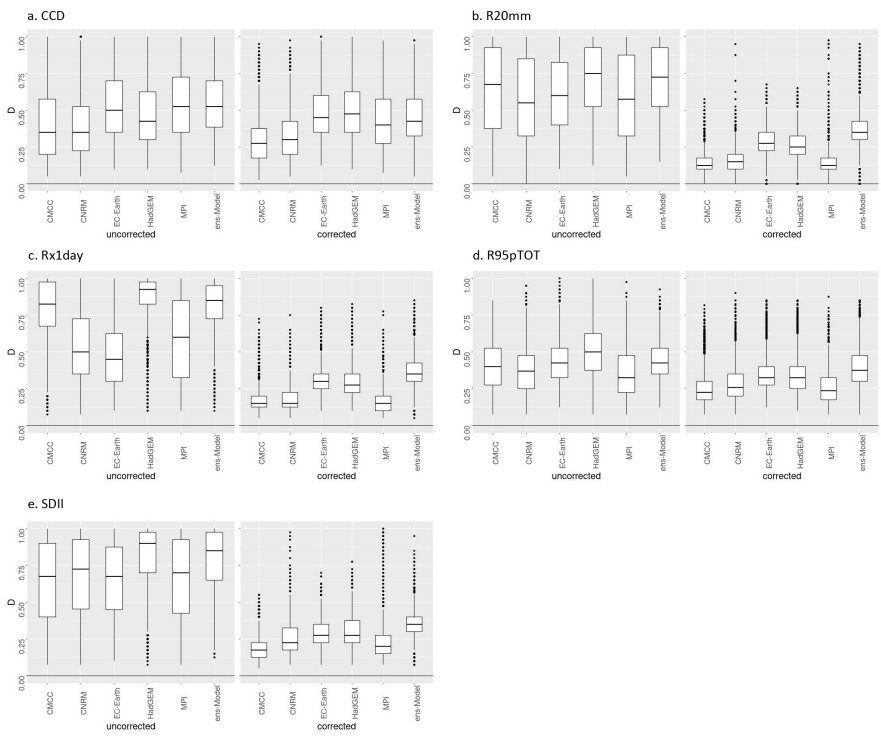

**Figure 2.** The Kolmogorov–Smirnov statistic value between the original model dataset and bias-corrected model dataset. Distance value (D) on the vertical axis shows the distance between the cumulative distribution of the observation and the model. The D value was calculated for simulating a) CDD, b) R20mm, c) Rx1day, d) R95pTOT and e) SDII.

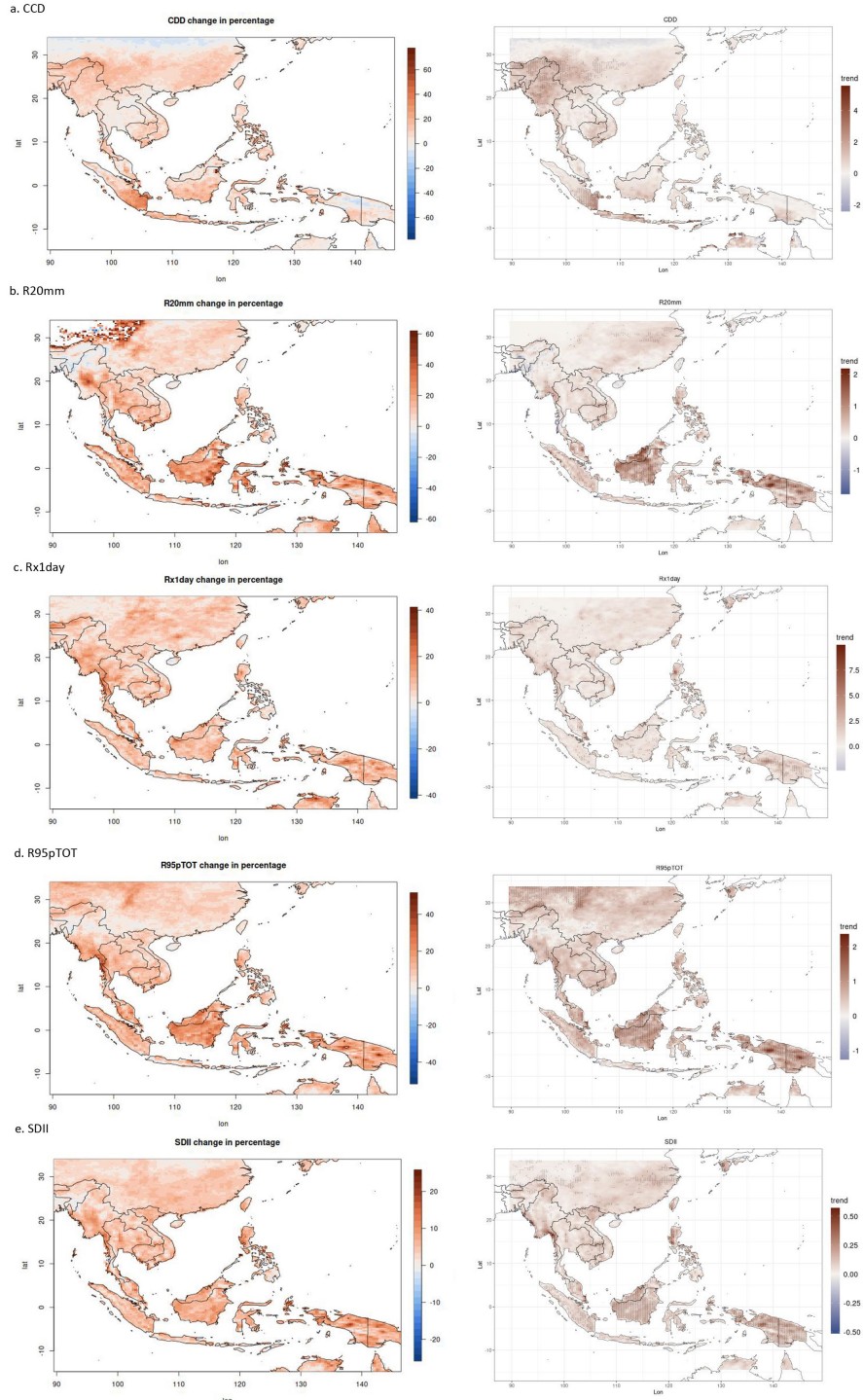

**Figure 3.** The change in percentage of the near future (2021-2050) compared to the historical period (1981-2010) (left), and the trend period 1971-2050 (right) for annually a) maximum length of a dry spell (CDD), b) a number of very heavy rainfall (R20mm), c) maximum daily rainfall (Rx1day), d) precipitation percent due to R95p days (R95pTOT) and e) simple daily intensity index (SDII). The dashes in the trend map indicate model agreement in the trend significant at 60% level agreement.

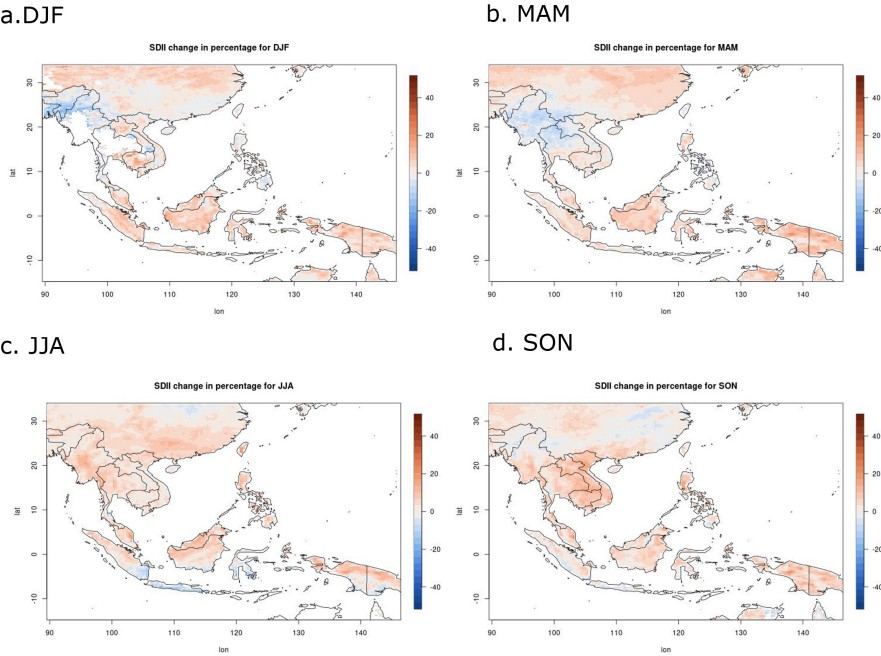

**Figure 4.** Seasonal Simple daily intensity index (SDII) change in percentage in the near future (2021-2050) compared to the historical period (1981-2010).

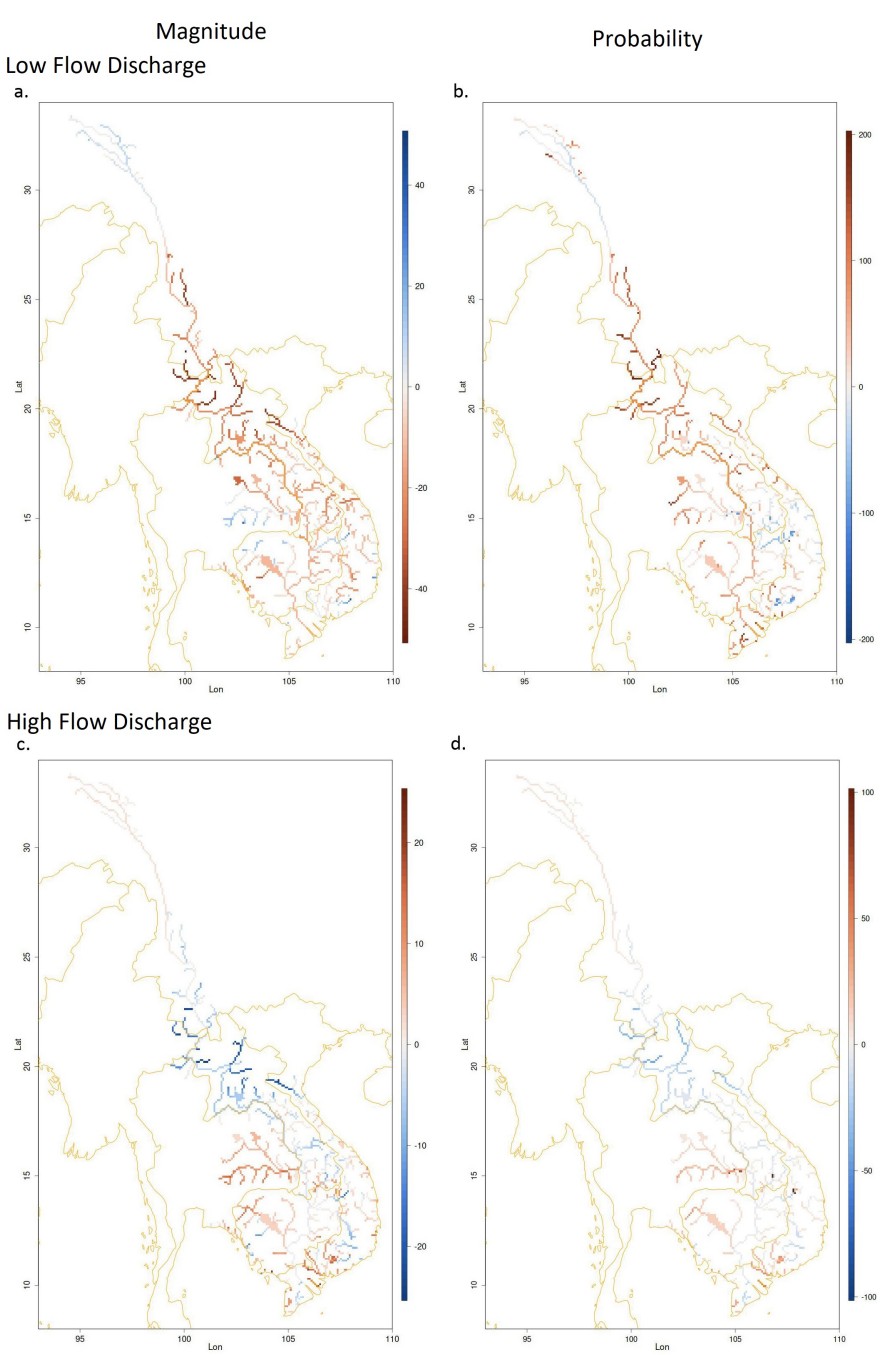

**Figure 5.** The change of extreme low (percentile 10) and extreme high (percentile 95) water discharge in the near future (2021-2050) compared to the historical period (1981-2010) over the Mekong region. Figure a: low flow magnitude change (%), figure b: low flow probability change (%), Figure c: high flow magnitude change (%) and figure d: high flow probability change (%).

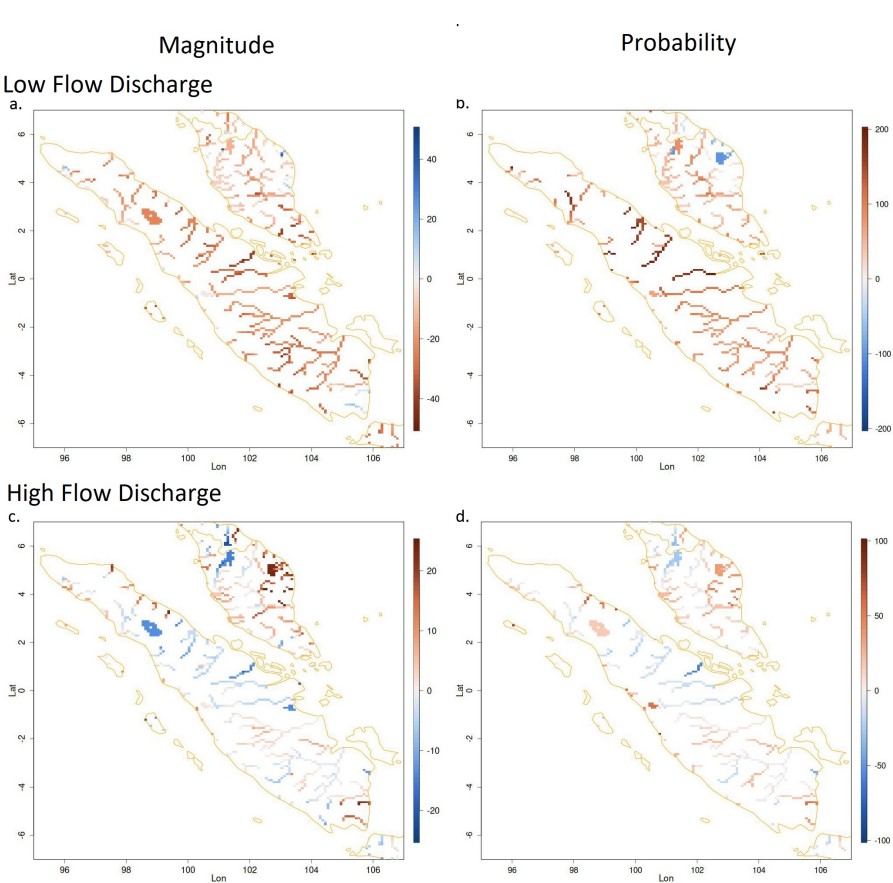

**Figure 6.** Similar to Fig. 5, but now for Sumatra and Peninsular Malaysia region.

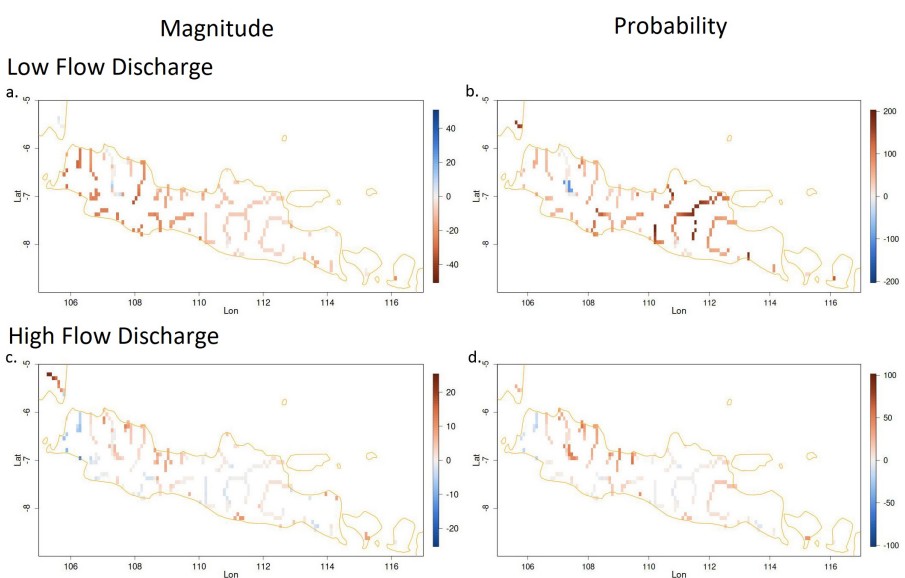

**Figure 7.** Similar to Fig. 5, but now for the Java island.

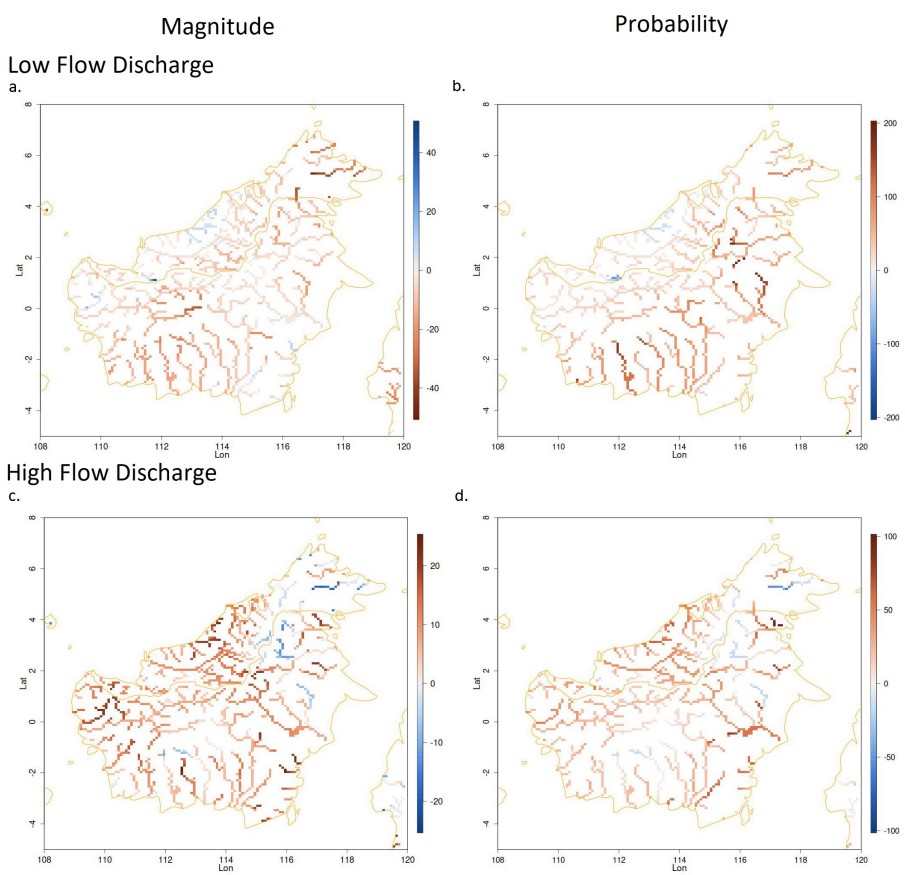

**Figure 8.** Similar to Fig. 5, but now for the Borneo island.

**Table 1.** List of rainfall related extreme climate indices computed in this study. The indices are calculated annually.

| Index ID | Index name | Index definition | Unit |
|---|---|---|---|
| CDD | Maximum length of dry spell | The largest number of consecutive days where rainfall is less than 1mm | d |
| CDD5D* | number of CDD >5 days | Number of CDD periods with more than 5days per time period | n |
| CWD | Maximum length of wet spell | The largest number of consecutive days where rainfall is at least 1mm | d |
| CDW5D* | Number of CWD >5 days | Number of CWD periods with more than 5days per time period | n |
| R10mm | Number of heavy rainfall days | The number of days where rainfall is at least 10 mm | d |
| R20mm | Number of very heavy rainfall days | the number of days where rainfall is at least 20 mm | d |
| Rx1day | Maximum daily rainfall | Highest one day precipitation amount | mm |
| Rx5day | Maximum 5-days rainfall | Highest five day precipitation amount | mm |
| R5day50mm* | Number of 5 days heavy precipitation periods | The number of 5 day periods with precipitation totals greater than 50 mm | n |
| R95pTOT | Precipitation percent due to R95p days | The ratio of the cumulative rainfall at wet days with $RR > RR95$percentile to the total rainfall | % |
| SDII | Simple daily intensity index per time period | The ratio of annual total rainfall to the number of wet days | mm/day |

*\* not derived from the ETCCDI climate indices*