# Peer review of "A high-resolution perspective of extreme rainfall and river flow under extreme climate change in Southeast Asia"

_Hydrology and Earth System Sciences, 2023_

## Author Response (AR1)

Dear Editor,

I am writing in response to your comments based on the reviewer's comments for our manuscript titled "A high-resolution perspective of extreme rainfall and river flow under extreme climate change in Southeast Asia" submitted to Hydrology and Earth System Sciences. We appreciate the opportunity to revise our paper based on the insightful comments provided by you and the reviewers. In this letter, we address each of your concern and outline the revisions we have made accordingly. In addition, In the track changes version, we have coloured the changes in red and used strikethrough for the deleted sentences

Your comment:

*RC1 #4 regarding uncertainties: There should be three main sources of uncertainties when quantifying climate change on streamflows. They are climate forcing inputs, hydrological model structure, hydrological model parameters. Other uncertainties can include water demand but that is not relevant when modelling natural streams or when you are assuming no anthropogenic interference. Your manuscript has discussed only uncertainties coming from the forcing inputs and hydrological model structure. The equifinality of hydrological model parameters is another important source of uncertainty.*

Our response:

*We sincerely appreciate your and the reviewer's valuable input regarding uncertainties in the quantification of climate change impacts on streamflow. We acknowledge that our initial manuscript discussed only uncertainties arising from climate forcing inputs and hydrological model structure, while neglecting the importance of hydrological model parameters as another significant source of uncertainty. In response, we have classified the uncertainty to three sources of uncertainty and added the uncertainty related to the hydrological model parameterization. The used of Hamon method for potential evapotranspiration estimation that might amplify the effect of changing temperature on the result. In addition, the parameterization of deep groundwater, soil layer and land cover also contribute to the uncertainty of the flow simulation (**P15-16L477-498**).*

Your comment:

*RC2 #2 We do agree with the reviewer that means rainfall and extremes are very seasonally dependent. However, in this paper, we would like to focus on the general changes in extreme events for both rainfall and streamflow. RC2 made a valid point here that seasonal changes are important to consider and would add value to your manuscript.*

Our response:

*We acknowledge your and the reviewer's comment highlighting the importance of considering seasonal changes in extreme rainfall. The motivation to use annual extremes is that we want to assess the changes in the annual lowest and annual highest extreme events as these have the largest impacts on society. However, we do agree with the reviewer that mean rainfall and extremes are very seasonally dependent. Therefore, we added seasonally analysis of extreme rainfall in the paper. The figures of seasonally change for rainfall indices are available in **Supplementary material 8-12**, and the result are explained in **P10L297-310**.*

Your comment:

*RC2 #5. You stated: In addition, previous studies by Trambauer et al., (2014) and Ward et al., (2013) show that PCR-GLOBWB is reliable for extreme studies.*

- *Trambauer et al., (2014) Comparison of different evaporation estimates over the African continent does not seem to show PCR-GLOBWB is reliable for extreme studies.*
- *Ward, P., Dettinger, M., Jongman, B., Kummu, M., Sperna Weiland, F., and Winsemius, H.: Flood risk assessment at the global scale-the role of climate variability, in: EGU General Assembly Conference Abstracts, pp. EGU2013–1390, 2013*

*This is an abstract and insufficient to support your argument here. Please provide strong evidence to show that PCR-GLOBWB is reliable for extreme studies.*

Our response:

*We would like to express our gratitude to you for bringing to our attention the insufficiency of the provided references to support our claim regarding the reliability of PCR-GLOBWB for extreme studies. We apologize for the oversight and any confusion caused. In response, we have removed the unsupported reference and replaced it to:*

- *Van der Wiel, K., Wanders, N., Selten, F., and Bierkens, M.: Added value of large ensemble simulations for assessing extreme river discharge in a 2 C warmer world, Geophysical Research Letters, 46, 2093–2102, 2019.*
- *Candogan Yossef, N., Van Beek, L., Kwadijk, J., and Bierkens, M.: Assessment of the potential forecasting skill of a global hydrological model in reproducing the occurrence of monthly flow extremes, Hydrology and Earth System Sciences, 16, 4233–4246, 2012.*

*We also added reference that show the used of PCR-GLOBWB for Sumatra Island in Indonesia from;*

*Meng, Y., Liu, J., Leduc, S., Mesfun, S., Kraxner, F., Mao, G., Qi, W., and Wang, Z.: Hydropower production benefits more from 1.5 C than 2C climate scenario, Water.*

Once again, we sincerely appreciate you time and efforts in evaluating our manuscript. We believe that the revisions we have made adequately address the concerns raised, resulting in an improved version of our paper. We are confident that the updated manuscript now meets the high standards set by Hydrology and Earth System Sciences and contributes valuable insights to the field of climate change impact on streamflow.

Thank you for considering our revised manuscript. We look forward to hearing from you soon.

Sincerely,

Mugni H Hariadi

Corresponding authors

Dear Reviewer 1

We would like to express our gratitude for taking the time to review our manuscript titled "A high-resolution perspective of extreme rainfall and river flow under extreme climate change in Southeast Asia" and for providing valuable feedback. We appreciate your insightful comments and suggestions for improvement. We would like to point out that this manuscript is not intended to test the performance of the PCR-GLOBWB hydrological model but we used the model as a tool to simulate the streamflow. This manuscript focused on the impact of extreme climate change (RCP 8.5) on extreme rainfall and furthermore on the low and high streamflow. The model validations are available from the previous paper by Sutanudjaja et al. (2018). In addition, previous studies by Van der Wiel et al. (2019) and Candogan Yossef et al. (2012) show that PCR-GLOBWB is reliable for extreme studies.

Furthermore, in response to your comments, we have carefully revised the manuscript to address the concerns raised. We address your comment point by point in the table below, **P** refers to page and **L** refers to line number, e.g., **P1L9** means page 1 line 9.

Major comment:

| No | Comment | Response |
|---|---|---|
| 1 | The message conveyed by the authors was not clear, does extreme rainfall influence changes in low and high flows, or increased low flow under climate change scenario? Or something else. I suggest the author to better rephrase it throughout. | We understand the reviewer concern about the message of the paper. However, we stated our main finding in the abstract "the impact of climate change is more prominent in a low flow event than in a high flow" **(P1L9-10)**. Moreover, we also discuss this in the Section 4.2, e.g., "the impact of changes in climate indices in SEA also affects the changes in hydrological extremes. The increase of CDD over Northern MEB results in declining LFD over the central part of MEB" **(P13L418-419)**. We will change the title of the sub-chapter for clarification. The title of section 4.2 change from "Extreme climatic changes" to "The impact of changes in climate indices to the hydrological extremes" (**P13L407**). In addition, we have added brief description about the content of the sessions in the last paragraph of the introduction. |
| 2 | The research gap being addressed in the current form seemed too weak and not clear. Para 2 and 5 in the introduction may summarized previous facts. But it still not clears why the research is important to do. Also, does it any benefit using CMIP for the analysis compared to other datasets. | We sincerely appreciate for the feedbacks. We understand that the reviewer concern about the unclear message of paragraphs 2 and 5. To make it clearer, we added these points below:
• Sentences that mentioned most of previous studies that investigated the effects of climate change on hydrological system were mainly based on CMIP5 models instead of CMIP6 models
• Previous studies show that CMIP6 has better representation of the physical, chemical, and biological processes than CMIP5. |

| | | | • A reference that demonstrates if the HighResMIP (High Resolution CMIP6) models generate more accurate monsoon characteristic and extreme rainfall over SEA compared to downscaling result of CMIP5 (CORDEX) (Hariadi et al. (2021,2022)).
Those points are written down in Paragraph 5 (P3L71-86). |
|---|---|
| 3 | It seemed that the authors would like to deliver the message on the importance of catchment properties to flows dynamics under climate change scenario (L15). But it may not be supported by strong methods and findings. Is there any calibration/validation on the hydrological model? If yes, which stations were used for this? Findings in Figs 3-6 may not show the changes in low/high flow, but it indicates the changes in water depth/storage in the river networks. River discharge is measured in fixed station not along the river network. | The importance of catchment properties to streamflow changes due to climate change is not the main message of our paper. The objective of our paper is to evaluate the changes in future extreme rainfall and streamflow. However, we found that not all river basins in SEA follow the changes in extreme rainfall and our result suggests that the catchment properties/memory may play a significant role here. Therefore, we discuss the importance of catchment properties in the discussion. We have revised our manuscript accordingly thus the main message of our is clearly stated and the discussion on catchment properties will not overcome the main message. We remove the last statement in the abstract that mention "Our study highlights the importance of catchment properties in aggregating and/or buffering the impact of extreme climate change" and add information about the discussion based on river in Java and Sumatra in **P1L13-15**.
We used the river recession constant ($C$) to indicate the catchment properties. The method and the map of recession constant are available in the supplement material (**Supplementary 13**). The validation of the PCR-GLOBWB hydrological model is available in the previous paper by Sutanudjaja et al. (2018) and in Section 2.2.2 (**P6L157-161**). The river discharge (streamflow) is calculated using the kinematic wave for the routing method (**P5L153**), which allows flow and area to vary both spatially and temporally within a conduit. Thus, the PCR-GLOBWB simulates the river discharge in $m^3/s$ for all river networks (**P6L154-155**). |
| 4 | I appreciate that authors mentioned about uncertainty of the work. My question is how the uncertainty influences the conclusion/findings. Better elaborate on it will improve the readability of the manuscript | We have elaborated the uncertainty influences the in the manuscript **(P15-16L477-498).**
In addition, we have classified the uncertainty to three sources of uncertainty. We also have added the uncertainty related to the hydrological model parameterization. The used of Hamon method for potential evapotranspiration estimation that might |

| | | amplify the effect of changing temperature on the result. In addition, the parameterization of deep groundwater, soil layer and land cover also contribute to the uncertainty of the flow simulation. |
|---|---|---|
| 5 | Please elaborate what PCR-GLOBWB model can do and how does it work? Are there any assumptions for the model. | We have added more explanations of the PCR-GLOBWB model in chapter 2.2.2. The PCR-GLOBWB is essentially a leakybucket type of model [Bergström, 1976] applied on a cell-by-cell basis. Daily for each grid cell, PCR-GLOBWB calculates the water storage in two vertically stacked soil layers (max. depth 0.3 and 1.2 m) and an underlying groundwater layer, as well as the water exchange between the layers and between the top layer and the atmosphere (rainfall, evaporation and snow melt). The explanation is in **P5L137-141.** |
| 6 | **L131-137** not part of data, I suggest the authors to provide new sub-section about statistical analysis | **L131-137** in the previous version is the validation of the PCR-GLOBWB hydrological model summarized from the previous paper (Sutanudjaja et al., 2018). We will make it clear in the manuscript that the detailed validation is available from previous paper and it is publicly available (**P6L157)**. |
| 7 | Hydrological drought was firstly mentioned in L55, which may not coherence with the previous paragraph. Please revise to better connect with the previous one. Does hydrological drought become a focus of study? I think is not as not many supports afterwards. | We thank for the useful feedback. Indeed, the reviewer is correct that this study analyzes the low flow and not necessarily drought. We revised it by removing the first sentence (**L55** in the previous version) for better connection between paragraphs 3 and 4. |
| 8 | L386-395 shall be in the method section to represent the catchment properties influence. Also, how steep or shallow hydrograph was determined was not found anywhere in the method. | **P13L386-395** in the previous version is a brief information for the river recession constant. The detail method on how to calculate the river recession constant is presented in the supplement material (**Supplementary 13**). |

Minor comment:

| No | Comment | Response |
|---|---|---|
| 1 | L2 does any climate change is not extreme? | We refer the extreme climate change to RCP 8.5 |
| 2 | L9 more prominent, how much? | We used prominent to highlight that the change of low flow is higher compared to high flow. The value is available in the chapter 3.2. |
| 3 | L129 the authors used Hamon method for calculating potential ET may need elaboration (e.g. why not other methods?) the method may not commonly used n Southeast Asia. | We only used rainfall and temperature in the analysis, this limit us to used more advance method like the Penman Monteith method. In addition, the Hamon method is available in the PCR-GLOBWB. We have add the information on the manuscript (**P5L152**) |
| 4 | L145 aggregated? | The number of the event is cumulated per year. |
| 5 | L158 put a comma after Indonesia | Thank you. We have added the comma. |

| 6 | L534 PhD thesis? | It is a master thesis. We have revised the manuscript. |
|---|---|---|

We sincerely appreciate your thorough evaluation of our manuscript and your constructive feedback. We believe that the revisions made have significantly enhanced the manuscript's quality, scientific rigor, and relevance. We are confident that these improvements position our study as a valuable contribution to the field of hydrology and climate change research.

Thank you for considering our revised manuscript for publication in HESS journal.

Sincerely,

Mugni H Hariadi

Corresponding authors

Dear Reviewer 2

We would like to express our gratitude for your time and consideration of our manuscript titled "A high-resolution perspective of extreme rainfall and river flow under extreme climate change in Southeast Asia" in the context of HighResMIP CMIP6 models and the projection of extreme rainfall and river flow in Southeast Asia under future climate change. We appreciate your valuable comments and concerns, and we have taken them into careful consideration during the revision process. Below, we address each of your concerns and outline the actions we have taken to address them adequately, **P** refers to page and **L** refers to line number, e.g., **P1L9** means page 1 line 9.

Major comment

| No | Comment | Response |
|----|---------|----------|
| 1 | I think the literature review was not broad enough. As far as I know CORDEX – SEA has carried out multimodels downscaling projection over Southeast Asia. In fact, according to one of their papers (Tangang et al. 2020-- C*limate Dynamics,55, 1247-1267*), the projected future changes are consistent with what described in this study. However, I find there was lack of discussion on the comparison of the two simulations i.e. CORDEX-SEA vs HighRESMIP. | The comparison of CORDEX-SEA vs HighRESMIP are conducted in our previous paper (Hariadi et. al., 2021 and 2022). We will elaborate more on the comparison between CORDEX-SEA vs HighResMIP in the revised version. (**P3L74-78**) |
| 2 | The mean rainfall and extremes are very seasonally dependent. Yet, this study only considers annual extremes. I propose the authors to consider the changes in different seasons in addition to annual timescale. | The motivation to use annual extremes is that we want to assess the changes in the annual lowest and annual highest extreme events as these have the largest impacts on society.
However, we do agree with the reviewer that mean rainfall and extremes are very seasonally dependent. Therefore, we added seasonally analysis of extreme rainfall in the paper. The figures of seasonally change for rainfall indices are available in **Supplementary material 8-12**, and the result are explained in **P10L297-310.** |
| 3 | This study appears to have lack of model validations. I think it is useful to validate the HighResMIP models in terms of their ability in simulating the seasonal mean climate, including the monsoonal circulations. | The performance of the HighResMIP model and the comparison between monsoon characteristic and rainfall indices are available in the previous paper (Hariadi et. al., 2021 and 2022). It is mentioned that the CMIP6 high-resolution Modeling Intercomparison Project (HighResMIP) (Haarsma et al.,2016) result simulated closer monsoon characteristics and rainfall indices to observation than CMIP5 downscaled result from CORDEX-SEA. We are summarizing the findings in the introduction (**P3L74-78**). |

| 4 | The study used bias-corrected outputs to evaluate the projected changes. Yet, there was no analysis (other than Fig 1) on how bias-correction changes the future projected as compared to the raw outputs. It seems from a recently published paper (Ngai et al. 2022 -- Weather and Climate Extremes, 37, 100484) inconsistency in the magnitude and direction of future change can occur between the bias-corrected and the raw outputs of CORDEX-SEA simulations. The authors need to provide information how bias-correction can affect the projected climate change signals in HighResMIP models. | We thank the reviewer for the feedback. We used the same bias correction method as Ngai et al. (2022), so the impact of bias correction on the simulations can be expected to be significant, as Ngai et al. (2022) showed. In addition, the objective of the paper is to document the changes in hydrology. Quantifying the impact of bias correction is out the scope of this study. Furthermore, we referred Ngai et al. (2022) to show the impact of bias correction on the simulations in the revised manuscript. (**P7L195-204**) |
|---|---|---|
| 5 | I also find lack of validation in the river flow simulation during the historical period. I think this is needed before the models can be confidently used for projections. | The detailed validation of the PCR-GLOBWB hydrological model is available in the previous paper by Sutanudjaja et al. (2018). We refer this in **P6L157-161**. In addition, previous studies by Van der Wiel et al. (2019) and Candogan Yossef et al. (2012) show that PCR-GLOBWB is reliable for extreme studies(**P6L163-166**). |

Minor comment:

*I suggest the authors to use "Malay Peninsula" instead of "Malaysian Peninsula". Alternatively "Peninsular Malaysia" can be used if the authors was referring to the west Malaysia. However, if the landmasses they are referring to include the southern Thailand then "Malay Peninsula" would be most appropriate.*

Response

*We thank the reviewer for the feedback. We agree with the reviewer and thus we are now using "Peninsular Malaysia" for referring the west Malaysia in the revised manuscript.*

We sincerely appreciate your thorough evaluation of our manuscript and your constructive feedback. We believe that the revisions made have significantly enhanced the manuscript's quality, scientific rigor, and relevance. We are confident that these improvements position our study as a valuable contribution to the field of hydrology and climate change research.

Thank you for considering our revised manuscript for publication in HESS journal.

Sincerely,

Mugni H Hariadi

Corresponding authors

---

## Author Response (AR2)

Dear Reviewer

We would like to express our gratitude for dedicating time to review our manuscript, titled "A high-resolution perspective of extreme rainfall and river flow under extreme climate change in Southeast Asia" and for providing valuable feedback. Your insightful comments and suggestions for improvement have been greatly appreciated. Furthermore, in response to your comments, we have carefully revised the manuscript to address the concerns raised. We address your comment point by point in the table below, **P** refers to page and **L** refers to line number in the track-change version, e.g., **P1L9** means page 1 line 9.

Comment 1.

*The message conveyed in the introduction is not clear enough. In particular, the science question is well described, which however is not connected to the motivation very well. Based on my reading, it seems that the motivation is to use CMIP6, as a successor of CMIP5 and CORDEX-SEA, to investigate the impact of climate change, although this science question has been addressed previously and extensively. This gives me the impression that this study is a modeling study instead, in which I am expecting to see more analysis on the improved performance of CMIP6 over its predecessors. I suggest the authors either think about adding new experiments (maybe in specific regions) to compare the results of using different climate forcing, or elaborate the motivation in a more clarified and concise way.*

Response 1.

*We acknowledge the reviewer's concern regarding the message in the introduction. After receiving feedback from the reviewer, we realize that paragraph 5 may mislead readers about the objective of the study. The focus of our study is to investigate the impact of climate change on streamflow, rather than to compare the performance between HighResMIP and CORDEX-SEA. In response to this comment, we have removed paragraph 5. We have also elaborated some references to the next paragraph (P3-4L87-101).*

Comment 2.

*There lacks a validation on the hydrological simulations. I understand that the model was previously validated in the same regions. Are those simulations using the same forcing as your study? It would be helpful to clarify this, as the hydrological simulation depends significantly on the atmospheric forcing being used.*

Response 2.

*The simulation for the validation in the previous study used different forcing compared to those employed in this study. We have added an explanation about the forcing that is used for the simulation in the validation, "Sutanudjaja et al. (2018) validated the PCR-GLOBWB 2.0 simulation with streamflow data from the Global Runoff Data Centre (GRDC). The forcing data set for the simulation is based on time series of monthly precipitation, temperature, and reference evaporation from the CRU TS 3.2 (Harris et al., 2014) downscaled to daily values with ERA40 (1958–1978) (Uppala et al., 2005) and ERA-Interim (1979–2015) (Dee et al., 2011)" (P6L172-175).*

Comment 3.

*In the discussion (especially in the section 4.1), I appreciate the authors' thorough analysis against other available studies in the same region. When making comparison, it may be even better to explain what factors may cause the difference between your results and others'. What are the relevant uncertainties? There is a separate section on the sources of uncertainty. But that is more like a general discussion.*

Response 3.

*There are two differences between HighResMIP used in this study and the CORDEX-SEA employed in the previous study by the CORDEX group. Firstly, HighResMIP is based on the CMIP6 version, whereas CORDEX-SEA is based on CMIP5. Secondly, HighResMIP is globally run at a high resolution, whereas CORDEX-SEA is derived from downscaling results from CMIP5, and thus a more indirect product. Previous studies have indicated that HighResMIP simulation shows better performance than CORDEX-SEA simulation. To address the reviewer's comment, we have added this information in P13L397-405.*

Comment 4.

*Figures need reorganization.*

| NO | Comment | Response |
|---|---|---|
| 1 | *Specially, in figure 1, are there statistical quantities (in addition to whisker plot) that you can use to better demonstrate the improvement from the bias correction?* | *We have added additional information about the statistical index in the caption. The information is "Distance value (D) on the vertical axis shows the distance between the cumulative distribution of the observation and the model".* |
| 2 | *In the second column of figure 2, the range of color maps need to be adjusted to improve readability.* | *We have revised the legend.* |
| 3 | *Some figures relevant to the discussion (such as those in section 4.3) need to be moved from the supplement to the main context* | *The analysis of recession constant in section 4.3 is used as supplementary support for the finding in the result section. While integral to the overall study, it does not constitute the main goal of the study, therefore we keep the figure in the supplementary material.* |
| 4 | *Line 297, It is more appropriate to either move the figures relevant to the discussion here or move the discussion to* | *Thanks for the suggestion. We have moved the figure of SDII in* |

| | *supplement.* | *the main manuscript (Figure 4)* |
|---|---|---|
| 5 | *In section 2.1, a map with some texts showing geographic regions is recommended.* | *Thanks for the suggestion. We have added figure of the Southeast Asia Domain in figure 1.* |

**Minor comments**

| No | Minor comment | Response |
|---|---|---|
| 1 | Section 2.2.1, when describing the difference among climate data (e.g., CORDEX-SEA and HighResMIP), it may be good to mention their corresponding resolution. | We have added the resolution information in P5L131-132 |
| 2 | "historically forced atmosphere run of HighResMIP (HighResSST), the Hist-1950" What is the difference? A bit more elaboration is helpful. | The HighResSST is the HighResMIP experiment with prescribed SST, whereas the Hist-1950 is the HIghResMIP experiment with atmosphere-ocean couple model.

We have added this information in P5L136-138 |
| 3 | Line 125, "high skill" is weird wording and is not accurate. | We have removed "*high*" |
| 4 | Line 131, It is actually possible for GCMs to simulate at high resolutions. So it may be good to delete "what is possible". | We have removed "*what is possible*" (P5L147) |
| 5 | Line 148, "The model runs in 5 arcmins spatial resolution, which is about 10 km by 10 km at the equator." This does not seem to be a high resolution for GHMs as claimed early in this paragraph. | The global hydrological model (GHM) simulates distributed hydrological responses to climate and weather variations at a higher resolution than in general circulation models (GCMs). In addition, for a global hydrological model 5 arcmins spatial resolution is relatively high resolution.

To clarify this we address this comment with removing "what is possible" (the same with |

| | | previous comment P5L147) |
|---|---|---|
| 6 | Line 150, What remapping method do you use? | We used the First-order conservative remapping method. We have added the information in P6L165. |
| 7 | Line 168-173, This paragraph is not necessary. Or you may want to move it upper | We have removed this paragraph |
| 8 | Line 200, "DJF". This is defined later in page 10. | We have added the definition (P7L218-219) |
| 9 | Line 222, "is" -> does | We have changed "*is*" to "*does*" (P8L241) |
| 10 | Line 247, How is the probability calculated? Section 2.3.3 "The probability change is calculated based on the change of the number of extreme low and extreme high events in the future compared to the historical" | We have added the information about how the probability calculated in P9L268-269. "*The probability change is calculated based on the change of the number of events that exceeded extreme low and extreme high reference values in the future compared to the historical.*" |
| 11 | Line 256, Please rewrite this sentence to improve clarity. | We have rewritten it to "Acknowledging uncertainties, where extreme values from certain models are included in the averaging process, a trend significance test was also performed. This test is grounded in model agreement and remains unaffected by extreme values, as they are excluded 280 from the trend analysis." (P9L277-280) |
| 12 | Line 321, This explanation is unnecessary for the readers of HESS. | We have removed it |
| 13 | Line 323, The change of probability can be up to 200%? Same question for figure 4-6. It is useful to define how the probability is calculated more clearly in the methodology section. | We have added the information about how the probability calculated in P9L268-269. "*The probability change is calculated based on the change of the* |

|  |  | *number of events that exceeded extreme low and extreme high reference value in the future compared to the historical*". For example, if the occurrence of future extreme is doubled than the historical period, then the probability increase is 100%. |
|---|---|---|

We sincerely appreciate your thorough review of our manuscript and the constructive feedback provided. We believe that the revisions have significantly enhanced the manuscript's quality, scientific rigor, and relevance. We are confident that these improvements position our study as a valuable contribution to the field of hydrology and climate change research.

Thank you for considering our revised manuscript for publication in HESS journal.

Sincerely,

Mugni H Hariadi

Corresponding author

---

## Author Response (AR3)

Dear Reviewer

We are writing to express our gratitude for your time and effort in reviewing our manuscript, titled "A high-resolution perspective of extreme rainfall and river flow under extreme climate change in Southeast Asia". Your insightful comments and suggestions for improvement have been greatly appreciated. Furthermore, in response to your comments, we have carefully revised the manuscript to address the concerns raised. We address your comment point by point in the table below.

| No | Comment | Response |
|---|---|---|
| 1 | L14-15: shallow hydrographs
It is better to use 'flat hydrographs' in contrast to 'steep' or flashy hydrograph
Please ensure that you are consistent with the terminology throughout the manuscript. You used different terms such as steep-slope (e.g. L554) or flashy hydrograph, gentle hydrographic (e.g. L489), shallow-slope hydrograph etc. Please correct all of them to be consistent. | We have revised the manuscript. L14-15 shallow change to flat. L453-455 flashy and gentle change to steep and flat respectively, L520-L521 remove slope and changed shallow to flat.
L522 change shallow-slope to flat hydrograph |
| 2 | L17: The IPCC's sixth assessment report (IPCC6AR)
It is best to write IPCC 6AR (with a space in between) or IPCC AR6 | We have changed it to IPCC AR6 in L17 |
| 3 | L104: constructed to document the changes in climate indices, in this case; rainfall indices
Change to 'constructed to document the changes in the rainfall-related climate indices'. It will make the text consistent with your Section 2.3.1 | We have changed "constructed to document the changes in climate indices" to "constructed to document the changes in the rainfall-related climate indices" in L79-80 |
| 4 | L280: This test is grounded in model agreement
Change to This test is based on the model agreement | We have changed "This test is grounded in model agreement" to "This test is based on the model agreement" in L248 |
| 5 | L400: In this subchapter
Change to: In this sub-section | We have changed "subchapter" to "sub-section" in L367 |

We believe that the revisions have strengthened the manuscript. We look forward to receiving your final feedback. Thank you for your time, we truly appreciate your contribution to our work.

Thank you for considering our revised manuscript for publication in HESS journal.

Sincerely,

Mugni H Hariadi

Corresponding author